# Towards Practical Large-Scale Privacy-Preserving Recurrent Neural Networks

## Abstract

Recurrent neural networks (RNNs) are used for a variety of applications such as speech recognition and financial forecasting where data privacy is an ongoing concern. Fully homomorphic encryption (FHE) facilitates computation over encrypted data, enabling third-party services like machine learning inference while keeping client data private. Previous studies have examined RNN inference over encrypted data using FHE, albeit on a small scale, though impractical due to the computational costs. This work advances insights that make large-scale RNN evaluation over encrypted data practical. A problem that prohibits the scaling of privacy-preserving RNNs is overflow in the ciphertext message space. As the number of model parameters increases, the size of the domain during multiply-accumulate operations increases, causing inaccuracies in computation. Attempts to mitigate this problem, such as splitting the message into several ciphertexts, cause an exponential increase in computation, making latency-sensitive applications like RNNs impractical. A novel regularization technique is proposed that mitigates the effects of numerical overflow during training. This allows use of one ciphertext only and reduces the complexity of the encryption parameters that would otherwise be required to perform correct computation while maintaining 128-bit security. Using the CGGI variant of FHE and GPU acceleration, we quantize and evaluate a 1.9M parameter, multi-layer RNN across 28 timesteps, achieving 90.82% top-1 accuracy over the encrypted MNIST test dataset with an average latency of 2.1s per sample—a new state of the art in latency, model performance, and scale.

## 1 Introduction

Machine learning as a service powers many important applications, from advanced recommendation systems to speech recognition, by computing over data. While privacy can be maintained during transport and storage, any service that performs computations on data exposes it in clear text, creating a potential vulnerability. Fully homomorphic encryption (FHE) [Gentry, 2009] provides a solution by enabling computation over encrypted data, thereby safeguarding data privacy during computation.

There has been extensive research into evaluating different types of neural network models over encrypted data using FHE [Podschwadt et al., 2022]. However, there have only been a few investigations into recurrent neural networks (RNNs) [Lou & Jiang, 2019; Podschwadt & Takabi, 2021; Anonymous, 2025], that can be attributed to two characteristics of FHE: (i) ciphertexts contain noise which increases with every consecutive mathematical operation, and (ii) FHE operations are computationally expensive. As noise grows with every consecutive operation, it eventually reaches a point where the message in the ciphertext becomes corrupted, unless an operation referred to as *bootstrapping* is performed that reduces the noise. RNNs have a variable and often very large depth in the time dimension, making them more susceptible to ciphertext corruption than other traditional networks, thus necessitating the use of bootstrapping to unlock unlimited depth. Due to the significant computational overhead inherent in FHE operations and large number of operations in variable-length RNNs, latency is significant, rendering even modest networks impractical due to inefficiency.

This work centers on scaling RNN evaluation over encrypted data to handle larger and deeper networks while simultaneously maintaining model performance and decreasing inference latency. We evaluate RNNs in a *non-interactive* client-server setting where once data is encrypted and sent to

the server, the client does not perform any other operation on the data until it receives the final result for decryption. The server applies all necessary mathematical FHE operations on the encrypted data.

Since one of our focuses is on reducing latency, this work employs the use of the CGGI [Chillotti et al., 2020a] variant of FHE. The CGGI scheme provides a latency-efficient bootstrapping operation, referred to as programmable bootstrapping (PBS), that can be accelerated by GPUs [Zama, 2022]. The PBS operation can evaluate any function that can be represented by a lookup table, which includes non-linear activation functions, while simultaneously reducing noise. However, CGGI can only perform operations over signed integers within a bounded domain, which results in the need for RNN quantization. We adopt the four-step RNN quantization procedure from Anonymous [2025], which quantizes RNNs into a ternary parameter and binary activation representation. This method allows us to represent a single activation using one ciphertext rather than several, reducing the required computation by orders of magnitude. While [Anonymous, 2025] is effective in reducing computational cost, it demonstrates decreased model performance for the following reason: since activations are signed integers, multiply-accumulate operations, which result in pre-activations, can overflow the modulus of the message space. This causes inaccuracies after activation function evaluation and a resulting drop in performance.

In response, this work bridges this gap by introducing a novel regularization method during training in order to mitigate overflow and increase model performance. **Overflow-aware activity regularization (OAR)** pushes pre-activations to *correct overflow regions*. For instance, due to the use of the `sign` function as the sole activation function in the network, a single overflow causes positive pre-activations to become negative—an incorrect result. However, if the pre-activation is trained to overflow one more time, the activation becomes positive—a correct result.

Utilizing a 1.9M parameter, multi-layered RNN architecture alongside GPU acceleration, this work attains a noteworthy 90.82% top-1 accuracy when tested on the encrypted MNIST test dataset, exhibiting a marginal deviation of $-0.17\%$ from plaintext performance, coupled with an average latency of $2.1\,\mathrm{s}$. Notably, this outcome showcases a latency reduction of 274x when compared to SHE [Lou & Jiang, 2019], alongside negligible disparities in accuracy between plaintext and encrypted runs, despite a 10x increase in model parameters and 3x augmentation in layer count. The implementation of OAR significantly boosts model performance, with certain configurations demonstrating a nearly 71% increase in top-1 accuracy when compared to non-OAR counterparts. Visual examination of pre-activation distribution histograms confirms that OAR effectively guides values to regions that result in accurate behavior. Furthermore, analysis of the average error between activations during encrypted and plaintext executions reveals minimal discrepancies.

In section 2, relevant background information is reviewed. In section 3, overflow-aware activity regularization is introduced. Section 4 presents and analyzes experimental results and provides a comparison to the literature in section 5.

## 2 Preliminaries

### 2.1 Fully Homomorphic Encryption

Fully homomorphic encryption (FHE), first proposed by Gentry [2009], is a form of encryption that enables the evaluation of mathematical operations over encrypted data. For example, if two ciphertexts that encrypt a value of "1" are added together, the result would be an encryption of "2". The security of modern FHE constructions is based on the Learning With Errors (LWE) problem [Regev, 2005] in lattice cryptography which adds a small noise sample to the message during encryption. As such, operations on FHE ciphertexts increases overall noise magnitude with each consecutive mathematical operation. In decryption, a rounding operation is performed that removes the noise added during encryption. During this step, if the noise is too large in magnitude, the message can be corrupted. A ciphertext can undergo a *bootstrapping* operation before the noise reaches corruptible levels, revealing the ability to perform an unbounded number of operations on each ciphertext [Marcolla et al., 2022], important to RNN evaluation. FHE schemes are initialized by a security parameter $\lambda$ used to generate encryption parameters and keys. As the parameters are increased, the amount of computation required to perform FHE operations increases as well. A value of $\lambda = 128$ is used throughout this paper [Marcolla et al., 2022].

## 2.2 THE CGGI SCHEME AND PROGRAMMABLE BOOTSTRAPPING

The CGGI [Chillotti et al., 2020a] mathematical foundation of FHE operates over the real torus. Its discretized version can encode and encrypt integers $x \in \mathbb{Z}_k$, where $k = 2^\omega$ is the plaintext modulus and $\omega$ is the bit-width. As a result, operations occur over the set of signed integers modulo $k$, generating a message space $x \in [-k/2, k/2)$. CGGI offers the ability to perform (1) addition between two ciphertexts, (2) addition between a plaintext and ciphertext, and (3) multiplication between a plaintext and ciphertext. Since the message space is bounded, operations can cause the message in the ciphertext to overflow and wrap around the modulus, a characteristic important to this work. For instance, if $x \in [-8, 8)$ and $k = 2^4$, then $7 + 1 = -8$ rather than 8.

CGGI enables simultaneous evaluation of discrete functions and noise reduction of ciphertexts through programmable bootstrapping (PBS). With encryption parameter $N$ (the ring dimension) being the number of elements in lookup tables (LUTs) and a power of two, a function $f(x)$ can be encoded into a LUT by dividing its elements into $k/2$ sub-packs of $N/(k/2)$ elements, each set to $f(x)$, where $x \in [0, k/2)$. Each sub-pack corresponds to a specific value of $x$, sequentially, ensuring tight coupling between each value in $x$ and its corresponding sub-pack. This facilitates the evaluation of $y = f(x)$ for any ciphertext input. For instance, consider the `sign` function,

$$\text{sign}(x) = \begin{cases} +1 & \text{if } x \geq 0, \\ -1 & \text{otherwise.} \end{cases} \tag{1}$$

A LUT with all elements equal to 1 can evaluate the `sign` function. Despite the absence of negative values in the LUT, the negacyclic property implicitly encodes $2N$ values, where the second set of $N$ values evaluates to the negation of the first set during PBS evaluation. Noise in the message can lead to incorrect output values, particularly if the encryption parameters are insufficient, especially for $x$ values close to 0 and $k/2 - 1$, which may overflow due to the negacyclic property. Utilizing a smaller plaintext modulus without changing the encryption parameters increases the sub-pack size, enhancing the probability of correct decryption amidst noise but reducing the message space size. Hence, when encryption parameters persist, there exists an inversely proportional relationship between the message space size and the probability of correct decryption. Thus, if an efficient set of encryption parameters is selected, decreasing the plaintext modulus increases accuracy, a relationship key to our work.

## 2.3 QUANTIZATION OF RNNS

CGGI operates with bounded signed integers, while neural networks typically use floating-point, necessitating quantization of inputs, parameters, and activations [Bourse et al., 2018; Sanyal et al., 2018; Lou & Jiang, 2019; Chillotti et al., 2020b; Folkerts et al., 2023]. Quantization discretizes continuous domains and maps values to discrete elements [Liang et al., 2021], introducing noise equal to the difference between a quantized value and its original value. This noise impacts model performance, which is a key focus of research surrounding the quantization of neural networks [Gholami et al., 2022]. Quantization-aware training (QAT) is a popular technique that introduces noise during training to enhance model robustness [Jacob et al., 2018]. However, quantizing RNNs presents challenges due to the recurrence relationship between timesteps, leading to exploding values [Hou et al., 2021]. Techniques like (i) normalization after matrix multiplication and (ii) fixed-point quantization have shown promise [Hou et al., 2021; Hou & Kwok, 2018]. Regarding (i), normalization is unsupported by CGGI since it returns a quantized domain back to floating-point. Regarding (ii), although not directly supported by CGGI, fixed-point techniques can be adapted by encrypting each bit, and performing arithmetic operations over arrays of ciphertexts. Privacy-preserving neural networks using CGGI, such as those in [Lou & Jiang, 2019; Sanyal et al., 2018; Folkerts et al., 2023], leverage these techniques. Although adder and activation function circuits with multiple PBS operations can facilitate fixed-point arithmetic over an array of ciphertexts, the sheer number of FHE operations needed for these processes compared to a single addition or PBS operation is orders of magnitude higher. This exponential increase in computation and latency renders RNN evaluation impractical, nullifying any potential accuracy gains.

The technique in Anonymous [2025] successfully quantizes RNNs into binarized activations ($\{-1, 1\}$) and ternarized parameters/inputs ($\{-1, 0, 1\}$). Binarization in CGGI is performed by a PBS that uses a lookup table encoding the `sign` function (eq. 1). The four-step quantization algorithm in [Anonymous, 2025] (algorithm 1 in appendix A.1) creates an RNN where addition between integers

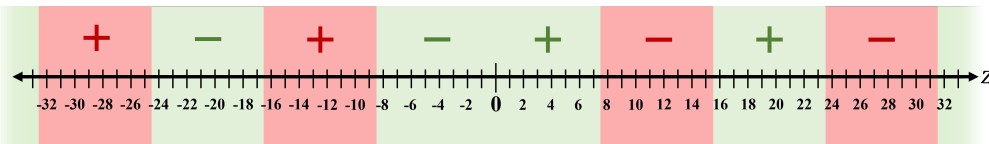

Figure 1: Correct (green/light shade) and incorrect (red/dark shade) outputs of the `sign` function (eq. 1) in $\mathbb{Z}_{16}$. "+" or "-" signs denote its output for values within the respective shaded regions.

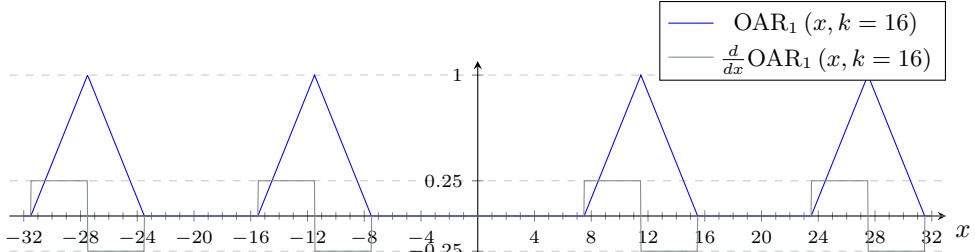

Figure 2: $\mathrm{OAR}_1(x, k = 16)$ and its derivative.

requires one addition operation and activation function evaluation requires one PBS operation, aiding in reducing latency significantly. In general, it utilizes QAT to quantize activations, inputs, and parameters in RNNs sequentially, in four steps. The algorithm leads to a quantized RNN with unbounded accumulation in matrix multiplication, increasing the likelihood of overflow in large networks. Accumulation emerges as a key challenge, prompting solutions like fixed-point arithmetic, as noted in works such as SHE [Lou & Jiang, 2019], which can manage accumulation of any size despite the computational overhead. In [Anonymous, 2025], the authors evaluate an RNN over encrypted data using 11 bits of precision, much larger than the 8-bit suggested maximum [Chillotti et al., 2021]. They mention that this causes overflow of the plaintext modulus, resulting in a 25% decline in top-1 accuracy over encrypted data when compared to plaintext. Therefore, this study delves into accumulation issues and proposes a regularization strategy to mitigate overflow, unlocking massive efficiency gains without compromising model performance.

## 3 OVERFLOW-AWARE ACTIVITY REGULARIZATION

Within perceptrons, the linear transformation of inputs $\boldsymbol{x} \in \mathbb{R}^n$ by parameters $\boldsymbol{W} \in \mathbb{R}^{m \times n}$ is represented by $\boldsymbol{z} = \boldsymbol{W} \cdot \boldsymbol{x}$ (bias vectors are omitted to align with the specific quantization strategy being employed [Anonymous, 2025]). This yields a "pre-activation" vector, which is then fed into an activation function to produce "activations". Pre-activations $\boldsymbol{z} \in \mathbb{R}^m$ form a distribution with a domain $[\alpha, \beta] \subset \mathbb{R}$. When weights are ternarized ($\boldsymbol{W} \in \{-1, 0, 1\}^{m \times n}$) and inputs are binarized ($\boldsymbol{x} \in \{-1, 1\}^n$), pre-activations become integers in $\mathbb{Z}$. Given a precision level $k = 2^\omega$ for bit-width $\omega$, which acts as a modulus, when operating in $\mathbb{Z}_k$ it is possible for values to overflow. By analyzing the effects of overflow in a neural network using this quantization representation, we make two observations.

**Observation 1: There are regions in $\mathbb{Z}$ where overflow in $\mathbb{Z}_k$ is inconsequential to the intended output of the `sign` activation function.** Figure 1 demonstrates the effect of overflow on the output of the `sign` function (eq. 1). In this example, consider a 4-bit modulus $k = 2^4 = 16$. Figure 1 displays a number line with values $z \in \mathbb{Z}$. The large sign symbols (+/-) within shaded regions represent the outputs of the `sign` function for values in that region, modulo $k$. Notably, some regions (red/dark shade) induce incorrect overflow in $\mathbb{Z}_k$, resulting in sign disparities between $\mathbb{Z}$ and $\mathbb{Z}_k$. For instance, within the region $[-16, -9]$, values have a sign of "-1" in $\mathbb{Z}$, but due to overflow, exhibit a sign of "+1" in $\mathbb{Z}_k$, leading to incorrect activation function outputs. Conversely, other regions (green/light shade) demonstrate correct overflow behavior, maintaining consistent signs between $\mathbb{Z}$ and $\mathbb{Z}_k$. For example, the value "18" retains a sign of "+1" in both sets, ensuring accurate activation function outputs despite overflow.

**Observation 2: Overflowing a value from an incorrect region into a correct one can flip the sign, aligning the `sign` activation function outputs across $\mathbb{Z}$ and $\mathbb{Z}_k$.** For instance, in figure 1, if a value such as "12", initially falling in the incorrect region $[8, 15]$, is relocated to the correct regions $[0, 7]$ or $[16, 23]$, the `sign` function outputs "+1" in both $\mathbb{Z}_k$ and $\mathbb{Z}$, reflecting the intended activation output. Based on this observation, since the values in figure 1 represent the pre-activations of a perceptron, we introduce a novel method that utilizes activity regularization to guide the network parameters in distributing pre-activations from incorrect regions to correct ones, ensuring consistent activation results with matching signs in both $\mathbb{Z}$ and $\mathbb{Z}_k$.

### 3.1 THE OVERFLOW-AWARE ACTIVITY REGULARIZER

The **Overflow-Aware Activity Regularizer (OAR)**, inspired by $\mathcal{L}_1$ regularization, penalizes values in incorrect regions based on their distance from the nearest correct region. The following set of equations define the novel OAR for a pre-activation input $x \in \mathbb{Z}$ and a modulus $k = 2^\omega$,

$$\text{OAR}_1(x, k) = \text{ReLU}\left( 1 - \frac{4}{k} \left| \left[ |x| - \frac{k-2}{4} \right]_{\text{mod } k} - \frac{k}{2} \right| \right) \tag{2}$$

$$\text{OAR}_2(x, k) = \text{OAR}_1^2(x, k) \tag{3}$$

To extend this to a loss function over pre-activation vector $\boldsymbol{x} \in \mathbb{Z}^n$, the minimization objective can be defined as,

$$\min_\vartheta \left[ \mathcal{L}_{OAR}(\boldsymbol{x}, k) = \sum_{i=0}^{n-1} \text{OAR}(x_i, k) \right] \tag{4}$$

where $\vartheta$ is the set of all parameters in the model, and OAR can be either equation 2 or 3.

In each incorrect region, the first half of values is closest to the preceding correct region, while the second half is closest to the succeeding correct region. Figure 2 depicts the $\text{OAR}_1$ regularizer for a modulus $k = 2^4 = 16$, with the penalty applied to each incorrect region represented by a *hat* function. The derivative function, also displayed, is positive for values needing adjustment to the preceding correct region and negative for those requiring adjustment to the succeeding correct region. Correct region values undergo no penalty, preventing unnecessary updates. Notably, OAR minimizes *strain* on the model by limiting the necessary movement of each incorrect value to a maximum of $k/4$ spots left or right. In this context, *strain* is defined as the effort a model exerts when changing parameters to accomplish an objective. In contrast, $\mathcal{L}_1$ or $\mathcal{L}_2$ regularization compels values to move an unbounded number of spaces towards zero, increasing strain on the model. This feature aids in balancing accuracy while minimizing parameter changes for optimal convergence. The *hat* functions are also translated by $0.5$ towards zero to maintain continuous derivatives at all points.

The OAR is employed in the fourth step of the 4-step quantization procedure outlined in [Anonymous, 2025] during quantization-aware training for every layer. Algorithm 1 extends the base algorithm with the OAR, marked in green as "NEW" (line 20). It is applied akin to general activity regularization, adding the OAR loss per layer (eq. 4) to the total model loss, with pre-activations serving as inputs. The OAR accommodates any modulus $k = 2^\omega$, where $\omega \geq 1$, facilitating generalization to any precision and integer quantization level. $\text{OAR}_2$ (eq. 3) closely resembles $\text{OAR}_1$, with parabolic *hat* functions and linear derivatives, imposing a higher penalty on values nearer to the center of incorrect regions, promoting faster convergence towards correct regions. Effectiveness is measured using the *OAR metric* which quantifies the percentage of pre-activations in correct regions relative to total pre-activations per layer.

### 3.2 SIGN ACTIVATION FUNCTION OVER A MODULUS

Algorithm 1 from [Anonymous, 2025] binarizes activations during quantization-aware training by applying the `sign` function (eq. 1) in the forward step while using the derivative of the `tanh` function during backpropagation. However, the `sign` function performed by CGGI operates over integers in $\mathbb{Z}_k$ for a modulus $k = 2^\omega$. To better mimic this environment, we propose converting pre-activations to their signed representation in $\mathbb{Z}_k$ before applying the `sign` function during training. In addition to the OAR, this would allow the model to learn from incorrect values resulting from

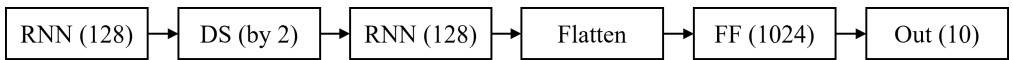

Figure 3: Architecture of MNIST RNN. DS (by 2) refers to downsample by a factor of 2, FF refers to feed-forward. Numbers in parentheses refer to number of units. Out is the output layer.

overflowing the modulus. The following function converts unsigned integers $x \in \mathbb{Z}$ to their signed representation $x_k \in \mathbb{Z}_k$ given a modulus $k = 2^\omega$,

$$x_k = \text{signed}(x, k) = \begin{cases} x \bmod k, & x \bmod k < k/2 \\ -(k - (x \bmod k)), & x \bmod k \geq k/2 \end{cases} \tag{5}$$

Equation 6 shows the proposed `ModSign` function for pre-activation $x \in \mathbb{Z}$ and a modulus of $k = 2^\omega$,

$$\text{ModSign}(x, k) = \text{sign}\left(\text{signed}(x, k)\right) \tag{6}$$

It is crucial that the `ModSign` function is applied alongside the OAR. Experimenting with the function itself, the model did not quantize and perform well. Thus, we propose changing the activation functions in the fourth step of the 4-step quantization algorithm [Anonymous, 2025] to the `ModSign` function that uses the derivative of `tanh` during backpropagation. These changes are reflected in algorithm 1, line 15

## 4 EXPERIMENTAL RESULTS

The goal of these experiments is to (1) evaluate the effectiveness of the Overflow-Aware Activity Regularizer (OAR), and (2) show that it allows us to execute large-scale RNNs over encrypted data more efficiently than previous works while maintaining model performance.

We utilize a large multi-layer RNN, akin to [Anonymous, 2025], on the MNIST dataset [LeCun et al., 2010] for handwritten digit classification on $28 \times 28$ pixel images. The MNIST RNN (figure 3) processes each row of the input image at every timestep, totaling 28 timesteps per classification, and comprising 1,914,368 parameters. Refer to appendix A.2 for more detail regarding the MNIST RNN. We train, quantize, and evaluate models in plaintext using TensorFlow [Abadi et al., 2016], QKeras [Coelho et al., 2021], and machine A from table 5 in appendix A.2. We perform evaluation over encrypted data using machine B (table 5), a modified version of the Concrete-Core library [Chillotti et al., 2020b] (with fixes, link in table 5) incorporating a CGGI [Chillotti et al., 2020a] implementation, and the threat model defined in appendix A.2.2. We adhere to a MNIST split of 57.5K training, 2.5K validation, and 10K test samples. In the following experiments, we conduct the fourth step of the 4-step procedure, which includes the OAR, thus initializing the same model from step 3 across experiments to obtain comparable results. For information regarding the first three steps, see appendix A.2. We use a learning rate of $10^{-5}$, gradient scaling temperature scale $s = 4$ for RNN layers, and a ternarization scale $t = 1.5$ for all layers, as required by algorithm 1, selected through a grid search. We experiment with the $\text{OAR}_2$ regularizer since it was found to be the most effective from an analysis comparing $\text{OAR}_1$, $\text{OAR}_2$, and $\mathcal{L}_2$ activity regularization (see appendix A.3).

### 4.1 OAR EVALUATION

#### 4.1.1 ACCURACY WITH AND WITHOUT OVERFLOW-AWARE ACTIVITY REGULARIZATION

We compare the test accuracy of models trained with and without OAR (settings A and B, respectively). Both settings use the `ModSign` activation function. Table 1 displays their accuracy results concerning the $\text{OAR}_2$ regularizer (eq. 3) for bit-widths up to 8, the practical limit for CGGI operations [Chillotti et al., 2021]. Each model undergoes 1000 epochs of training utilizing a regularization rate set at $10^{-3}$. The 'Difference' column in table 1, which displays the difference in test accuracy between settings A and B, shows minimal variation between both settings for bit-widths 7 and 8. Notably, the OAR metric, which measures the percentage of pre-activations positioned in the correct regions of the domain, remains high for both bit-widths, implying adequate coverage by integer precision. For both 5-bit and 6-bit, the OAR is crucial in obtaining high accuracy, with a +71% and +43% difference in accuracy when used. This is supported by a large positive difference in the OAR metric columns, showing that the OAR successfully moves pre-activations to the correct regions. Below 5-bit, both

Table 1: MNIST RNN test metrics with and without $OAR_2$ for different bit-widths.

| Bit-Width | Accuracy (w/o OAR) | Accuracy (w/ OAR) | Difference in Accuracy | OAR Metric (w/o OAR) | OAR Metric (w/ OAR) |
|---|---|---|---|---|---|
| 8 | 95.30% | 95.19% | -0.11% | 97.30% | 99.96% |
| 7 | 95.15% | 93.97% | -1.18% | 74.72% | 99.98% |
| 6 | 46.82% | 89.35% | **+42.53%** | 48.43% | 99.96% |
| 5 | 9.88% | 80.73% | **+70.85%** | 61.20% | 99.79% |
| 4 | 11.13% | 11.04% | -0.09% | 52.98% | 51.79% |
| 3 | 11.14% | 11.16% | +0.02% | 52.91% | 51.92% |

Table 2: MNIST RNN test accuracy with $OAR_2$ for bit-widths 5 and 6 at various regularization rates.

| OAR Rate | 5-bit Accuracy | 6-bit Accuracy | | OAR Rate | 5-bit Accuracy | 6-bit Accuracy |
|---|---|---|---|---|---|---|
| 0 | 9.88% | 46.82% | | $10^{-4}$ | 81.11% | 92.11% |
| $10^{-6}$ | 12.30% | 70.33% | | $10^{-3}$ | 80.73% | 89.35% |
| $10^{-5}$ | 19.77% | 87.58% | | $10^{-2}$ | 76.79% | 83.92% |

settings fail to surpass random accuracy levels, suggesting potential limitations of OAR in lower bit-widths, possibly due to excessively granular domain representations.

### 4.1.2 ACCURACY WITH DIFFERENT OAR REGULARIZATION RATES

Focusing on the optimal bit-widths of 5 and 6, we vary the regularization rates for $OAR_2$. Table 2 presents the MNIST RNN test accuracy for different $OAR_2$ rates (models trained for 1000 epochs). The test accuracy without OAR regularization is shown in the first row. Across both bit-widths, $10^{-4}$ yields the highest accuracy. Notably, the 5-bit setting exhibits greater sensitivity to rate changes, with accuracy shifting over 60% between $10^{-5}$ and $10^{-4}$. This sensitivity is expected due to the finer granularity of the 5-bit domain, where small fluctuations can lead to significant shifts between correct and incorrect regions. This observation is supported by the larger accuracy gap between the 6-bit and 5-bit settings at $10^{-4}$, with the former consistently outperforming the latter. Thus, both rate and bit-width are crucial hyperparameters in OAR regularization.

### 4.1.3 VISUALIZING PRE-ACTIVATION DISTRIBUTIONS

Section 4.1.1 confirms OAR's success in shifting pre-activations to correct overflow regions while maintaining high accuracy. In this section, we examine histograms of pre-activations in the FF(1024) layer throughout training. Figure 4 presents histograms for 5-bit and 6-bit iterations (from section 4.1.2) with $OAR_2$ regularization at a rate of $10^{-4}$. Dashed vertical lines mark +/- regions in $\mathbb{Z}_k$, with checkmarks and "x"-marks indicating correct and incorrect regions. In both graphs, the intended distribution division is clear, with large pre-activation amounts in correct regions and small amounts in incorrect ones. The 5-bit setting shows more granular regions compared to the 6-bit setting, with smaller distances between correct regions. Despite some spikes in incorrect regions, the distinction between regions remains clear, with larger spikes in correct areas. This supports the observations in section 4.1.2, potentially explaining the variance in test accuracy between 6-bit and 5-bit experiments.

### 4.1.4 OAR FOR LONGER AND LARGER RNNS

This experiment assesses OAR's impact on longer sequences and larger RNNs. Input images are resized from $28 \times 28$ to $128 \times 128$ pixels, increasing parameters to 8,480,768 and timesteps to 128. For the enlarged RNN, we initialized the learning rate cosine decay to $5 \cdot 10^{-6}$. Training utilizes 6-bit $OAR_2$ regularization at a rate of $10^{-4}$, yielding a test accuracy of 92.69%. This confirms OAR's efficacy for longer/larger networks, crucial for efficiency, especially considering the sequential evaluation nature of RNNs over extended timesteps. By facilitating the use of superior parameters for

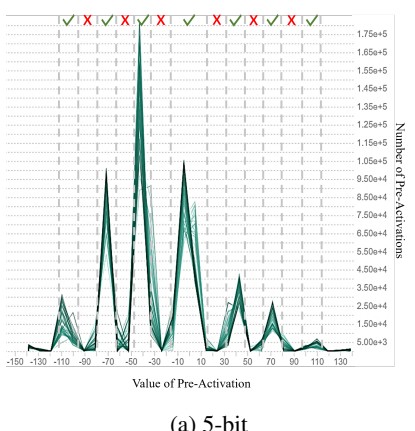

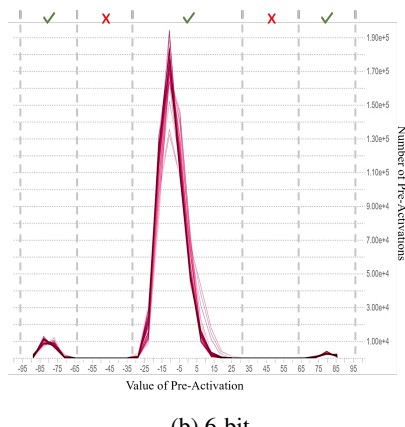

(a) 5-bit

(b) 6-bit

Figure 4: Pre-activation distribution histograms of the FF(1024) layer of the MNIST RNN, using 5-bit and 6-bit $OAR_2$ regularization with a rate of $10^{-4}$. Histograms extracted from TensorBoard.

faster efficiency without compromising accuracy, OAR proves crucial for optimizing performance in longer networks. Interestingly, the FF(1024) layer's OAR metric is 70.57%, suggesting complete pre-activation correction may not be necessary for good performance in larger networks. However, the network does not train without using both OAR and `ModSign`.

## 4.2 RNNS OVER ENCRYPTED DATA

We evaluate regular and enlarged MNIST RNNs over encrypted data using models from sections 4.1.2 and 4.1.4, trained with 6-bit $OAR_2$ at a rate of $10^{-4}$. Each model is tested with parameter sets 1 and 2 from table 6 in appendix A.2, ensuring a minimum security level of $\lambda \geq 128$ bits. Both sets support 6-bit message spaces, with set 1 providing higher precision in PBS operations but lower operational efficiency. Following appendix A.4, we accumulate pre-activations in the output layer into 4 ciphertexts per output unit to avoid overflow. `sign` activations are evaluated using a programmable bootstrap (PBS) and accelerated on two NVIDIA A100 GPUs, with each streaming multiprocessor evaluating one PBS in parallel. "encrypted" and "plaintext" denote evaluations over encrypted and plaintext data, respectively, using the 6-bit `ModSign` activation function for plaintext evaluations. Performance metrics include top-1 accuracy for encrypted and plaintext runs, and average latency. Error metrics include mean absolute error (MAE) between encrypted and plaintext pre-activations and percent difference (PD) between encrypted and plaintext activations, averaged across the test dataset. Standard deviation ($\sigma$) values are included for latency and error metrics to show average difference between samples, important for assessing the stability of our method since every sample is encrypted with normally-sampled FHE noise, potentially impacting metrics.

Top-1 accuracy and average latency results are presented in table 3 for both MNIST RNN models over the encrypted MNIST test dataset (10K samples). Table 4 displays PD and MAE metrics for each model, per layer. Using parameter set 1, the regular model achieves an encrypted top-1 accuracy of 90.86%, just 0.13% less than plaintext. The near-zero PDs in activations show negligible amount of error, indicating precise computation by the encrypted RNN. This is further supported by an output layer MAE between pre-activations of 2.02, significantly lower than the possible maximum of 1024, confirming minimal overflow impact due to effective overflow-aware activity regularization (OAR). In terms of efficiency, the model achieves an average latency of 4.86 seconds per sample. With parameter set 2, latency significantly decreases to 2.10 seconds per sample while maintaining accuracy within 0.17% of plaintext. Although parameter set 2 results in slightly higher PDs and MAE, the accuracy impact is negligible. The successful performance with a smaller parameter set highlights the effectiveness of OAR in reducing the need for large parameters and gaining efficiency as a result. Without OAR regularization, distributions would exceed the 8-bit modulus suggested for CGGI, requiring much larger parameters to maintain accuracy. Minimal $\sigma$ values indicate our method's stability despite the presence of FHE noise. Overall, these results demonstrate our approach's effectiveness in balancing accuracy and latency, enabled by our mitigation of overflow issues.

Table 3: Performance metrics for regular and enlarged MNIST RNN inference over encrypted data.

| Type of Model | Parameter Set ID | Encrypted Accuracy | Plaintext Accuracy | Difference in Accuracy | Average Latency (s) |
|---|---|---|---|---|---|
| Regular | 1 | 90.86% | 90.99% | -0.13% | $4.86 \pm 0.01$ |
|  | 2 | 90.82% | 90.99% | -0.17% | $2.10 \pm 0.01$ |
| Enlarged | 1 | 89.42% | 92.27% | -2.85% | $21.72 \pm 0.42$ |
|  | 2 | 87.56% | 92.27% | -4.71% | $10.26 \pm 0.03$ |

Table 4: Error metrics for regular and enlarged MNIST RNN inference over encrypted data, per layer.

| Type of Model | Parameter Set ID | First RNN Average PD (%) | Second RNN Average PD (%) | FF(1024) Average PD (%) | Out(10) MAE |
|---|---|---|---|---|---|
| Regular | 1 | $0.00 \pm 0.00$ | $0.10 \pm 0.14$ | $1.94 \pm 1.73$ | $2.02 \pm 1.31$ |
|  | 2 | $0.56 \pm 0.62$ | $0.46 \pm 0.29$ | $5.72 \pm 1.83$ | $3.96 \pm 1.37$ |
| Enlarged | 1 | $7.85 \pm 5.25$ | $11.42 \pm 3.28$ | $34.92 \pm 1.63$ | $53.06 \pm 13.82$ |
|  | 2 | $19.52 \pm 2.03$ | $20.80 \pm 1.79$ | $35.82 \pm 1.55$ | $53.38 \pm 13.92$ |

The enlarged model achieves excellent results for both parameter sets, despite a tenfold increase in error metrics across all layers, as shown in table 4. This sustained accuracy, despite higher error rates, demonstrates the robustness of the network due to overflow-aware activity regularization. As noted in section 4.1.4, the model failed to train without this regularization. The increased inaccuracies per layer are expected since each matrix multiplication involves nearly five times more multiply-add operations than the standard model, indicating an issue with FHE noise rather than overflow. This can be mitigated by including intermediary PBS operations to reduce noise. In terms of efficiency, the increase in latency is linearly proportional to the rise in timesteps (4.57x). The combination of strong encrypted accuracy and a linear latency response for a network that is significantly larger than the regular model highlights the effectiveness of OAR in scaling to larger and more extended RNNs.

## 5 RELATED WORK

The literature on non-interactive FHE evaluation of RNNs is limited to three key studies. (i) SHE [Lou & Jiang, 2019] uses multiple ciphertexts per input/activation, necessitating complex adder and activation circuits that are abundant in programmable bootstrapping operations, significantly reducing efficiency. Despite this, their fixed-point quantization approach achieves good accuracy, evaluating a single-layer RNN with 300 units over 25 timesteps on the Penn Treebank [Marcus et al., 1993] dataset. This setup, with approximately 180K parameters, completes one inference in 576 seconds, with a 2.1% accuracy drop from full-precision plaintext. Our implementation runs a 1.9M-parameter, multi-layer RNN across 28 timesteps in 2.1 seconds, achieving a 274x decrease in latency for a 10x increase in parameters, maintaining over 90% accuracy with an 8% drop from the 99% accuracy of full-precision plaintext MNIST evaluation. (ii) In another study, Podschwadt & Takabi [2021] leverage batched processing and eliminate quantization by employing the CKKS [Cheon et al., 2017] FHE scheme. Their method divides an RNN with $\tau$ timesteps into $n$ sub-RNNs, each with $\tau/n$ timesteps. While beneficial, this approach struggles with long-term dependencies, and the significant latency of one evaluation, at 19.5 minutes, makes it unsuitable for low-latency applications. (iii) A recent study by Anonymous [2025], which forms the basis of our research, presents a 4-step quantization procedure for CGGI evaluation of RNNs that allows the use of one ciphertext per input/activation for large efficiency gains. They evaluate a 12.6M parameter RNN with attention over encrypted data across 188 timesteps, achieving a latency of 531 seconds. Despite their excellent latency for such a large model with attention, there is a -25% accuracy drop between encrypted and plaintext top-1 accuracy due to using an 11-bit plaintext modulus. Our work introduces OAR to mitigate overflow effects from using a smaller modulus, recovering this lost accuracy while retaining the efficiency benefits of their quantization technique. Successful scaling to a larger RNN suggests that [Anonymous, 2025] may benefit from using OAR. Other works [Bourse et al., 2018; Sanyal et al.,

2018; Folkerts et al., 2023] evaluate different types of neural networks over the encrypted MNIST dataset using CGGI but do not investigate RNNs.

# 6 CONCLUSION

In this study, we proposed methods for non-interactively evaluating large-scale RNNs over encrypted data using fully homomorphic encryption. Employing a single-ciphertext representation for RNN inputs and activations is crucial for reducing inference latency, a major obstacle to scaling RNNs, but causes reduced model performance due to numeric overflow. Introducing overflow-aware activity regularization (OAR) effectively mitigates overflow effects and restores lost accuracy, demonstrating efficacy across RNN scales. Leveraging OAR and GPU acceleration, we evaluated a 1.9M-parameter, multi-layer RNN over encrypted MNIST, achieving remarkable results: 2.1 seconds latency and over 90% top-1 accuracy, setting a new state-of-the-art. Future research could optimize OAR further and explore deterministic methods for handling overflow regions in trained networks.

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

# A  APPENDIX

## A.1  MODIFIED FOUR-STEP QUANTIZATION ALGORITHM

---

**Algorithm 1:** Modified Four-Step Quantization Algorithm for RNNs from *Anonymous et al.* Anonymous [2025]

---

**Input**    : Vanilla RNN model $M_0$, trained or untrained, input dataset $\mathcal{D}$, ternarization scale $t$, **modulus $k = 2^\omega$ for $\omega$-bit accumulation**.

**Output** : Quantized RNN with binary activations and ternary inputs/parameters $M_4$.

---

**Step 1:**
1 Change activations in $M_0$ to tanh.
2 $M_1 \leftarrow \text{train}(M_0)$.

**Step 2:**
3 Change activations in $M_1$ to sign with tanh derivative.
4 $M_2 \leftarrow \text{train}(M_1)$.
5 **if** *exploding/vanishing gradients* **then**
6 ┊ Increase batch size.
7 ┊ Repeat lines 3 - 4.
8 **if** *model is still not training* **then**
9 ┊ **foreach** *RNN layer in $M_3$* **do**
10 ┊ ┊ Set temperature scale $s_l$ for gradient scaling.
11 ┊ Repeat line 3.
12 ┊ Repeat line 4 while applying gradient scaling.

**Step 3:**
13 $\mathcal{D} \leftarrow \text{ternary}(\mathcal{D}, \tau_I)$ where $\tau_I = t \cdot \mathbb{E}(|\mathcal{D}|)$.
14 $M_3 \leftarrow \text{train}(M_2)$.

**Step 4:**
15 **Change activations in $M_3$ to $\text{ModSign}(k)$ with tanh derivative (eq.6).**          /* NEW */
16 **foreach** *layer $l$ in $M_3$ w/ parameter distribution $\theta_l$* **do**
17 ┊ $\tau_l \leftarrow t \cdot \mathbb{E}(|\theta_l|)$
18 **foreach** *RNN layer $l$ in $M_3$* **do**
19 ┊ Set temperature scale $s_l$ for gradient scaling.
20 $M_4 \leftarrow \text{train}(M_3)$ while applying (1) gradient scaling, (2) ternarization in the forward step, **and (3) $\text{OAR}(k)$ in each layer.**          /* NEW */
21 **if** *model fails to train* **then**
22 ┊ Increase number of parameters in each layer of $M_0$ by $3x$.
23 ┊ Repeat lines 1 - 20.

---

## A.2 Detailed Experimental Setup

Table 5: Hardware and software experimental setup. Standalone numbers indicate corresponding version numbers. (x2) indicates there are two components.

|  | **Machine A** | **Machine B** |
|---|---|---|
| **CPU** | Intel i7-7820X, 8-Core, 3.60 GHz | AMD EPYC 7763, 64-Core, 3.1 GHz (x2) |
| **GPU** | Nvidia RTX 2080 Ti | Nvidia A100 40GB (x2) |
| **RAM** | 132 GB | 512 GB |
| **OS** | Ubuntu 22.04.2 LTS, 5.19.0-46-generic | Ubuntu 22.10, 5.19.0-46-generic |
| **CUDA** | 11.2.152 | 12.1.105 |
| **Python** | 3.10.6 | N/A |
| **TensorFlow** | 2.10.1 | N/A |
| **QKeras** | 0.9.0 | N/A |
| **Rust** | N/A | 1.69.0 |
| **Concrete-Core** | N/A | Link hidden due to double blind review. |

### A.2.1 Further Detail on the MNIST RNN and Training

As shown in figure 3, the first and third layers of the MNIST RNN consist of vanilla RNNs with 128 units. The second layer downsamples the outputs of the first RNN by half by stacking every two outputs, reducing the timesteps from 28 to 14, and increasing the size of the inputs to the third layer from 128 to 256. This temporal pooling aids in computation reduction while preserving accuracy [He et al., 2019]. The fourth layer flattens the outputs into a 1792-unit vector, connecting all RNN outputs to the classification layers. Following is a feed-forward network with 1024 units, culminating in an output layer of 10 units fed into a `softmax` function for probability generation. `tanh` activation is used in each layer as per the quantization strategy [Anonymous, 2025]. Bias parameters are omitted, as discussed in section 3. During training, the dataset is normalized by dividing by 255, and for enhanced generalization, we apply horizontal flips and random brightness adjustments to random images. The training dataset is randomly shuffled every epoch. Our training setup employs categorical cross-entropy loss, Adam optimization [Kingma & Ba, 2015], and a `cosine` schedule for the learning rate, decreasing to 0.1 times the initial value after 100 epochs. In the first three steps of the 4-step quantization procedure, we use a learning rate of $10^{-4}$, and a batch size of $512$. We train the first step for 100 epochs, and every subsequent step, including the fourth step, for 1000 epochs. In step one, we use the default parameter initializers of dense and RNN layers in TensorFlow (namely, "glorot_uniform" and "orthogonal" kernel and recurrent kernel initializers in RNN layers, respectively, and the "he_normal" kernel initializer in dense layers).

### A.2.2 Threat Model and Security Parameters

In this study, we embrace the "honest but curious" threat model, where participants conduct the privacy-preserving inference protocol with honesty but may seek insights from intermediate values. Through fully homomorphic encryption (FHE), security is maintained, encrypting all observed values and leveraging FHE's security guarantees [Rechberger & Walch, 2022]. Furthermore, this study evaluates neural networks that perform privacy-preserving inference over encrypted data, utilizing plaintext model parameters. Thus, FHE exclusively ensures the privacy of input data.

Bergerat et al. [2023] provide several sets of CGGI parameters that provide at least $\lambda = 128$ bits of security. Each set can accommodate different plaintext moduli $p = 2^{\omega}$, where omega is the bit-width. Sets with stronger parameters (i.e. larger $n$, $N$, and $k$ values) provide larger plaintext moduli but yield less efficient executions of CGGI algorithms. Refer to [Bergerat et al., 2023] for parameter definitions. We experiment with two sets of parameters, namely sets 1 and 2 in table 6, of which set 1 is sourced from row 5, table 4, and set 2 from row 12, table 8 in [Bergerat et al., 2023]. Since

the parameters in set 2 ($n$ and $N$) are smaller than in set 1, they result in smaller latency of CGGI operations. However, they support smaller plaintext moduli since the noise parameters, $\sigma^2_{\text{LWE}}$ and $\sigma^2_{\text{RLWE}}$, are larger, causing larger noise in ciphertexts. When compared to other sets, while they are rated for a 5-bit modulus, sets 1 and 2 are able to support larger multiply-accumulations (according to the $\nu$ metric which is proportional to the 2-norm of linear transformation vectors during dot-products, as defined in [Bergerat et al., 2023]). Thus, we are able to run experiments in a 6-bit message space, which was supported experimentally as well. Due to the smaller noise characteristics of set 1, it can handle more operations in a 6-bit message space than set 2. Consequently, the size of the dot-products that can be evaluated is less for set 2. Additionally, the ring dimension $N$ determines the size of lookup tables (LUTs) for evaluating functions using the programmable bootstrapping (PBS) operation, as defined in section 2.2. As the domain increases, each LUT sub-pack should increase in size as well. Parameter set 2 has a lower ring dimension, and thus, smaller sub-packs. As a result, it can cause more inaccuracies in PBS evaluation. For detailed information regarding definitions of the parameters in table 6, refer to [Chillotti et al., 2020a] or [Bergerat et al., 2023].

Table 6: CGGI security parameters ($\lambda = 128$) based on [Bergerat et al., 2023]. Please see [Bergerat et al., 2023] for parameter definitions. Select (*) parameters are modified for better precision while maintaining security guarantees. For both sets of parameters, $k = 1$, as defined in [Bergerat et al., 2023].

| Parameter Set ID | $n$ | $\sigma^2_{\text{LWE}}$ | $\log_2(N)$ | $\sigma^2_{\text{RLWE}}$ | $l_{\text{PBS}}$ | $\log_2(\beta_{\text{PBS}})$ | $l_{\text{KS}}$ | $\log_2(\beta_{\text{KS}})$ |
|---|---|---|---|---|---|---|---|---|
| 1 | 732 | $3.87088$ $(\times 10^{-11})$ | 11 | $4.90564$ $(\times 10^{-32})$ | 3 | 14 | 2 | 8 |
| 2 | 585 | $8.35721$ $(\times 10^{-9})$ | 10 | $8.93436$ $(\times 10^{-16})$ | 5* | 5* | 2 | 8 |

A.3   MNIST RNN ACCURACY WITH DIFFERENT ACTIVITY REGULARIZERS

This section evaluates the effect of different types of activity regularization on MNIST RNN accuracy over the MNIST test dataset. There are three different types of regularization defined in this work that can push pre-activations to the correct regions in $\mathbb{Z}_k$. $\mathcal{L}_2$ regularization can be applied to pre-activations to push them towards the first correct region, specifically $[k/2, k/2]$. $\mathrm{OAR}_1$ regularization pushes the pre-activations to all the correct regions, using a constant gradient for each value. $\mathrm{OAR}_2$ regularization is similar, except that it uses a gradient with magnitude relative to the distance of the value from the center of the incorrect region. Table 7 shows the test accuracy results for each regularizer's application over the MNIST RNN. The results in the table are recorded after training for 1000 epochs, using a precision setting of 6-bit and the `ModSign` activation function. From the table, it is evident that $\mathcal{L}_2$ regularization does not help increase the accuracy. However, for each run performed with $\mathcal{L}_2$ regularization, the OAR metric for each layer achieves almost 100%. Inspecting the pre-activation distributions reveals that the values are correctly moved to the region $[-32, 32)$, yet the accuracy does not increase. This observation, along with the better accuracy results for the runs with OAR, leads us to conclude that OAR regularization is necessary to both move values to correct regions *and* retain accuracy.

Table 7: MNIST RNN test accuracy with $\mathcal{L}_2$, $\mathrm{OAR}_1$, and $\mathrm{OAR}_2$ activity regularization. Precision of 6-bits is used.

| Regularization Rate | $\mathcal{L}_2$ | $\mathrm{OAR}_1$ | $\mathrm{OAR}_2$ |
|:---:|:---:|:---:|:---:|
| $1 \cdot 10^{-5}$ | 34% | 25% | 88% |
| $5 \cdot 10^{-5}$ | 31% | 25% | 91% |
| $1 \cdot 10^{-4}$ | 31% | 26% | 92% |
| $5 \cdot 10^{-4}$ | 30% | 44% | 92% |
| $1 \cdot 10^{-3}$ | 30% | 74% | 89% |
| $5 \cdot 10^{-3}$ | 31% | 69% | 86% |

Since we are using ternarization, a possible reason for the poor performance with $\mathcal{L}_2$ regularization is that most of the weights are pushed to zero, limiting the network's representational power. Effectively, this approach prunes the network, and the more the network is pruned, the lower the accuracy. In contrast, OAR regularization pulls weights towards zero and also pushes them outwards to other regions, causing less weights to be quantized as zero values. The number of quantized parameters that are zero divided by the total number of parameters is considered the pruning level. For instance, if this number is 80%, then 80% of the parameters in the model are zero. That being said, the pruning levels for the runs from table 7 with $\mathcal{L}_2$, $\mathrm{OAR}_1$, and $\mathrm{OAR}_2$ regularization, and a regularization rate of $10^{-3}$, are 84.96%, 77.58%, and 78.02% respectively. These numbers show that $\mathcal{L}_2$ regularization prunes the network more than OAR regularization (around 9% more in this case), suggesting that it could contribute to the observed lower accuracy.

Table 7 shows that $\mathrm{OAR}_1$ and $\mathrm{OAR}_2$ are effective regularization techniques for achieving high OAR metrics and accuracy. The table also shows that $\mathrm{OAR}_1$ is less effective than $\mathrm{OAR}_2$. In addition to the table, figure 5 shows the training curves of all the experiments in the table. On the right side of the table, the brackets aid in identifying the runs and their associated pre-activation regularization method. $\mathrm{OAR}_2$ consistently outperforms $\mathrm{OAR}_1$ in both the table and figure. The figure shows a better separation between the methods, with $\mathcal{L}_2$ at the bottom, $\mathrm{OAR}_1$ in the middle, and $\mathrm{OAR}_2$ at the top.

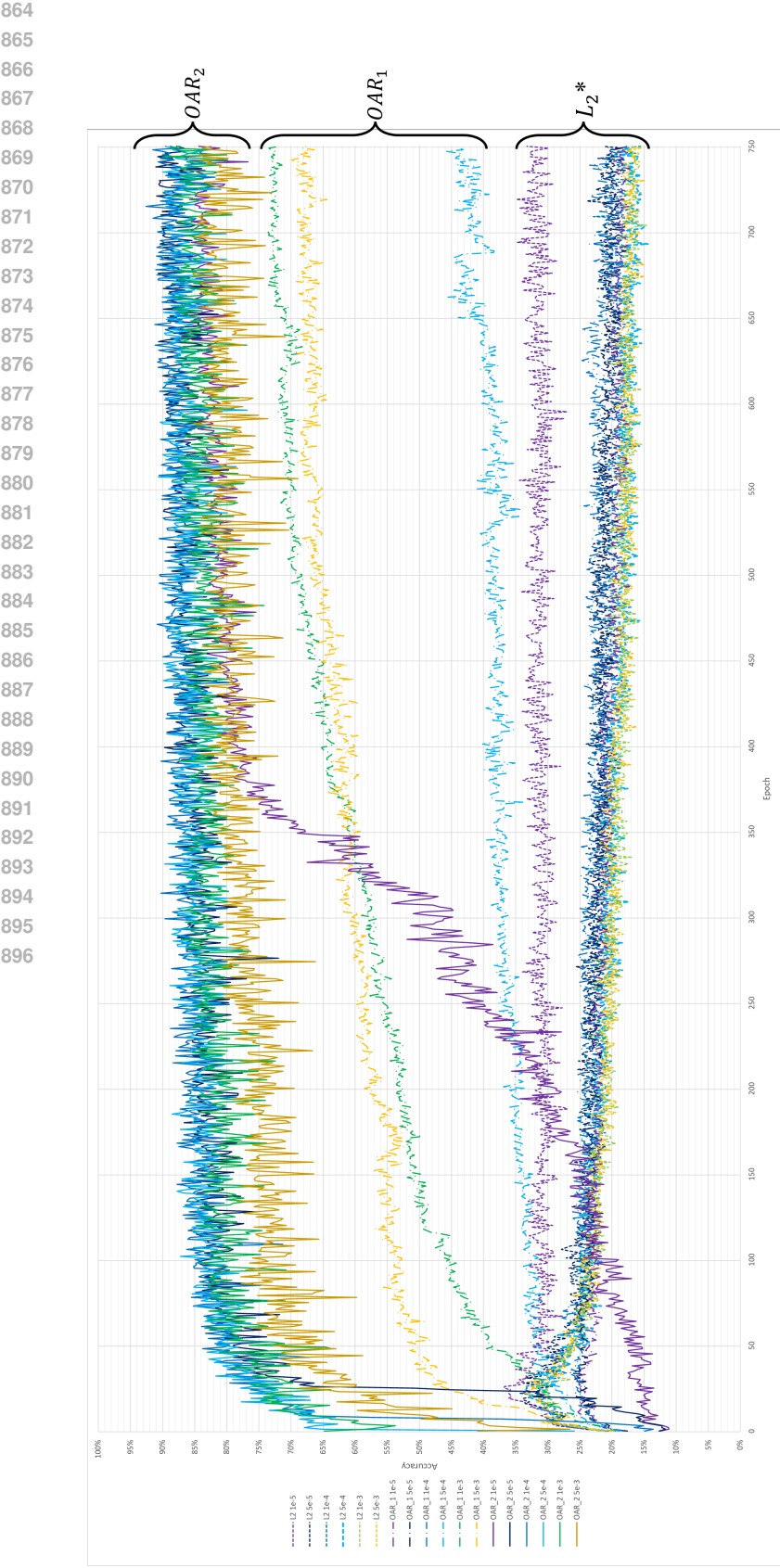

Figure 5: Training results comparing three different regularizers on the pre-activations of the MNIST RNN. OAR$_1$, OAR$_2$, and $\mathcal{L}_2$ are each used five times for different regularization rates. The metric displayed in the graph is the validation accuracy across the epochs. (*) In this region, there are also experiments with OAR$_1$ regularization. It is evident in the figure that OAR$_2$ performs the best with respect to accuracy, while both OAR regularizers perform much better than regular $\mathcal{L}_2$ regularization.

A.4 MANAGING OVERFLOW IN OUTPUT LOGITS

In the output layer of the MNIST RNN, pre-activations are fed into a `softmax` function. In this work, we do not calculate a `softmax` function in the output layer over encrypted data due to the difficulty of its implementation using FHE operations. Rather, we return the pre-activations to the client, who decrypts them and performs a `softmax` calculation in the clear. When calculating `softmax`, the relative magnitude between values is important for correctness. Therefore, if any of the pre-activations underwent overflow during matrix multiplication, `softmax` would generate an incorrect distribution. In section 3, we observed there is a way to harness overflow and make it useful. However, this observation depends on the use of the `sign` activation function (to reiterate, when overflow occurs, it is possible to overflow again and regain the intended sign of pre-activations).

We can increase the size of the CGGI security parameters to increase the available plaintext precision in the output ciphertexts, as suggested in appendix A.2, in order to support a larger accumulation space. In response, this would decrease the efficiency of the CGGI computation and not be desirable. Since the output layer is composed of a matrix multiplication operation only, the procedure in [Anonymous, 2025] divides the summation in each dot product into $d$ smaller summations, each summation represented by a ciphertext. Thus, $d$ ciphertexts represent each output unit. Instead of sending $o$ ciphertexts to the client for $o$ output units, this would send $o \cdot d$ ciphertexts. We do not worry about the expansion in network bandwidth since it is not a focus in this work. On the client side, the ciphertexts are decrypted and summed per output unit to complete the full summation.

As a result of the 4-step quantization process [Anonymous, 2025], the matrix multiplication operation between inputs $x \in \{-1, 1\}^i$ and parameters $W \in \{-1, 0, 1\}^{o \times i}$ in the output layer, where $i$ is the number of input units, is composed of a series of dot products between elements equal to "1" or "−1". This allows us to set an upper bound on the values of the output of each dot product. Consider $x \in \{1\}^i$ and $W \in \{1\}^{o \times i}$, then the maximum absolute value of dot products in the matrix multiplication is equal to $i$. Taking this into consideration, given a plaintext modulus $k$, $d$ can be adjusted such that the ciphertexts containing the smaller summations do not overflow—they would need to be able to handle pre-activations $z_k \in [-i/d, i/d]$, which requires a plaintext modulus $k \geq 2 \cdot i/d$. By increasing $d$, we can guarantee that there will be no overflow in any message space.

Equation 7 shows how the dot product that calculates each output unit $z_k$ can be divided into several smaller dot products. The dot product is between a row vector $w_k \subset W$ and $x$. It also shows how they can be recombined to evaluate the original dot product. In our RNN evaluation method with CGGI, each smaller dot product is calculated by the server through a series of additions or subtractions of binarized ciphertexts. The server sends every resulting ciphertext for each output unit to the client. The client decrypts the ciphertexts and recombines them according to equation 7 in the clear for each output unit.

$$z_k = \langle w_k, x \rangle = \sum_{j=0}^{i-1} w_{k,j} \cdot x_j = \underbrace{\sum_{j=0}^{i/d-1} w_{k,j} \cdot x_j}_{\text{Summation } 0} + \underbrace{\sum_{j=i/d}^{2i/d-1} w_{k,j} \cdot x_j}_{\text{Summation } 1} + ... + \underbrace{\sum_{j=(d-1)i/d}^{i-1} w_{k,j} \cdot x_j}_{\text{Summation } d-1} \quad (7)$$

A.5 LIMITATIONS OF OVERFLOW-AWARE ACTIVITY REGULARIZATION

In this study, we applied and evaluated OAR solely over the MNIST dataset. In future work, we will evaluate this method over other datasets to assess its generalization, which we were unable to perform due to time constraints. In section 4.1.1, we showed that the regularizer does not perform well for very low precision levels, such as 3 to 4 bit. It is possible that the pre-activation domain becomes too granular for the regularizer to have a meaningful impact, as suggested in sections of 4.1.1, 4.1.2, and 4.1.3. More research is required to investigate the application of OAR to lower precision levels. In appendix A.3, we showed the limitations of $\text{OAR}_1$ and how it did not perform as well. Future work will investigate the cause of this, possibly through a gradient analysis as the gradient flows from the output layer to the first RNN layer.

