# OpenReview forum: "Towards Practical Large-Scale Privacy-Preserving Recurrent Neural Networks"
_ICLR.cc/2025/Conference — Submitted to ICLR 2025_

### Official Review · Reviewer_4xG5 · 2024-10-28

**Soundness:** 1
**Presentation:** 2
**Contribution:** 2
**Rating:** 3
**Confidence:** 4

**Summary:**

This paper focuses on privacy-preserving RNN training using Fully Homomorphic Encryption (FHE). The authors build on their earlier work, adopting a four-step quantization procedure that ternarizes RNN parameters and binarizes activations to reduce computational costs. However, this quantization method comes with a trade-off: a reduction in model performance. The authors attribute this performance drop to overflow in the modulus of the message space during computation. To address this, they propose an overflow-aware activity regularization method, which aims to push pre-activation values into the correct overflow regions. The regularization works by penalizing pre-activation values that fall into incorrect regions, thereby forcing them into the appropriate range. The authors evaluate their method on the MNIST dataset and report minimal accuracy loss with the regularization applied.

**Strengths:**

- The Overflow-Aware Activity Regularization method that this paper proposes is novel.
- This paper achieves in much faster training speed than prior work.

**Weaknesses:**

- This paper builds upon prior, unpublished work by the same authors, which is provided as supplementary material. In the previous work, the authors propose a four-step quantization method that ternarizes model weights and binarizes activations to reduce computational costs and accelerate training. However, this prior work has not been peer-reviewed or rigorously examined. It only evaluates the quantization method on the VoxCeleb1 dataset (in plaintext), where the results indicate a significant drop in performance: top-1 and top-5 accuracy are reduced by 25% and 14%, respectively. Even in this paper’s evaluation on the MNIST dataset, the quantization method still shows a noticeable accuracy drop compared to full-precision training (from 99% to 90%). These concerns about the effectiveness of the prior approach cast doubt on the stability and reliability of the new method, which is built directly upon it.
- A key result claimed by this work is the improvement in training speed, purportedly achieving state-of-the-art performance. However, the speed gains primarily result from the quantization method introduced in the previous paper, not from the new contributions in this work. This paper's main innovation is the application of a regularization technique to stabilize training, but the method appears tightly coupled to the specific four-step quantization approach from the prior work. The authors provide no experimental evidence to suggest that the regularization method generalizes beyond this specific setup, which limits the broader applicability and impact of the paper.
- Additionally, the paper lacks sufficient experimental results to demonstrate the robustness of the proposed regularization technique. While the focus of the work is on private RNN training, the authors evaluate their approach solely on the MNIST dataset—a dataset designed for vision tasks. This raises concerns about the generalizability of the method to language-focused datasets, which are more relevant for RNN applications. Although the prior work was evaluated on the VoxCeleb1 dataset, this paper does not include any results on that dataset, further weakening the evidence for the method's effectiveness.

**Questions:**

I suggest the authors to either combine the two papers into a single one, or show that the regularization approaches generalize to a more general setting beyond the original paper, and provide thorough experimental evidence to support the claims.

---

> ### Author Response · Authors · 2024-11-25
> **Our work is centered on RNN inference over encrypted data, not training over encrypted data.**
>
> Thank you for your thoughtful and thorough review. We appreciate the time you've invested in evaluating our paper and would like to address your concerns point by point.
>
> ### Weakness 1:
>
> We believe there may have been a misunderstanding regarding the focus of our paper. Our work is centered on **RNN inference over encrypted data**, not training over encrypted data. While training is indeed a significant component, it is performed over plaintext data using quantization-aware techniques. Specifically, lines 253-254 in our paper state that the Overflow-Aware Regularization (OAR) is part of quantization-aware training on plaintext data. This process is essential to quantize the RNN so that it can be evaluated over encrypted data. Section 2.3 elaborates on the necessity of quantization for encrypted inference. Regarding the ICASSP paper, it is being peer-reviewed and we’ve received positive initial feedback.
>
> Regarding performance, we acknowledge that achieving 90.86% accuracy on the MNIST dataset may seem suboptimal compared to full-precision models. However, our primary aim is to demonstrate how OAR significantly **recovers the accuracy lost due to quantization**. As shown in Table 1, OAR provides substantial improvements over the baseline quantized models without OAR. For instance:
>
> - In the 5-bit setting, accuracy improves from **9.88% to 80.73%**, a gain of **70.85%**.
> - In the 6-bit setting, accuracy improves from **46.82% to 89.35%**, a gain of **42.53%**.
>
> These results highlight that OAR effectively mitigates the accuracy drop caused by quantization. When these models are evaluated over encrypted data—where the arithmetic operations match those during training—we can achieve these improved accuracies directly due to OAR. Figure 4 in our paper further illustrates that the pre-activations align with the desired distributions, confirming the effectiveness of our approach.
>
> ### Weakness 2:
>
> The key contribution of our paper is not solely an improvement in inference speed but the combination of **state-of-the-art latency and enhanced model accuracy** in the context of encrypted data. OAR is not applied to stabilize training; rather, it adjusts the pre-activation values to mitigate overflow issues inherent in homomorphic encryption computations. By correcting the pre-activation distributions, OAR ensures accurate inference over encrypted data without the need to split values into multiple ciphertexts, which would increase computational overhead.
>
> Our method builds upon the four-step quantization process because it is currently the **only known approach** that successfully quantizes an RNN to comply with the arithmetic constraints required for networks running over encrypted data using the CGGI framework, without splitting values into multiple ciphertexts. As such, the scope for generalization is limited within the current state of research. Our work is very specific in terms of the type of FHE scheme used and type of network considered. Anything outside of this is out of scope. Our paper focuses specifically on this method due to its suitability for encrypted inference without compromising on computational efficiency.
>
> ### Weakness 3:
>
> We chose to evaluate our method on the MNIST dataset and an **upscaled MNIST dataset** to demonstrate that our approach remains effective even when dealing with larger pre-activation distributions resulting from higher-resolution images. MNIST is a widely recognized benchmark in privacy-preserving machine learning (PPML), and many existing studies use it for initial evaluations. While we acknowledge that testing on language-focused datasets is important for RNN applications, the primary goal was to validate the efficacy of OAR in a controlled setting. Extending our evaluation to more complex and domain-specific datasets is a valuable direction for future work.
>
> **We want to stress that our work is a stepping stone towards the evaluation of large-scale RNNs over encrypted data, as is mentioned in the title. We do not claim anywhere in the work that this method will provide full-precision accuracy; rather, our claim is that it can help in certain low-bit situations and is a direction that should be explored and is worth it to be seen by the wider community.** Thank you again for your valuable feedback. We hope that our responses address your concerns and clarify the contributions of our work.

---

> > ### Comment · Reviewer_4xG5 · 2024-11-26
> >
> > Thanks you for your responses. Here are my comments:
> > * Again, I believe the contribution of this paper cannot be fully separated from your prior work. While the authors claim to focus on efficient and secure RNN inference, achieving state-of-the-art latency and enhanced model accuracy, the primary technique contributing to the state-of-the-art latency appears to originate from the prior paper. If the OAR technique proposed here is essential for the prior work to reliably achieve secure inference with high accuracy, this raises questions about the soundness and completeness of the prior work on its own. Only when the two papers are considered together can the full scope of the contributions to be fairly assessed.
> > * The authors claimed that OAR can help recover accuracy loss caused by quantization; however, evaluating this claim solely on MNIST is insufficient to validate its robustness. To demonstrate that OAR works under diverse circumstances, it should be evaluated on various common datasets typically used with RNNs. Showing accuracy recovery in a simplistic setting like MNIST does not guarantee similar performance on more challenging datasets. If the recovery does not generalize to harder datasets, the practical utility of the approach becomes limited.
> >
> > Once again, I suggest that the authors either combine the two papers into a single work or demonstrate that the proposed regularization approaches extend to a more general setting beyond the original paper. Additionally, thorough experimental evidence should be provided to substantiate the claims and validate the broader applicability of the methods.

---

> > > ### Author Response · Authors · 2024-12-01
> > >
> > > Thank you for your response! Please see our latest response to Reviewer qHBX. It also provides comments that are in response to your comments.

---

### Official Review · Reviewer_fiJU · 2024-10-31

**Soundness:** 3
**Presentation:** 2
**Contribution:** 2
**Rating:** 3
**Confidence:** 3

**Summary:**

This paper basically introduces overflow-aware activity regularization (OAR), a RNN regularization technique tailored for a usage within FHE setting. A parallel submission from the authors already proposed an efficient RNN quantization strategy, but with a decreased model performance as it might overflow the modulus of the message space. The authors experimented by showing a 1.9M parameter, multi-layered RNN architecture leading to 90.82% top-1 accuracy on MNIST, with an average latency of 2.1 s using FHE.

**Strengths:**

This work improves over another work submitted in parallel by the authors to another conference ICASSP 2025.

**Weaknesses:**

- I find it problematic that this submission extensively leverages some quantization techniques of another paper from the authors (Anonymous, 2025), submitted to another conference ICASSP 2025. Even though the paper's pdf is shared as an accompanying file, this does not belong to this submission. Yet, the quality and soundness of this paper can't be guaranteed, as the acceptance/rejection wasn't decided yet.  Also, this work proposes an improvement over the ICASSP 2025 paper, so why not simply update your results in that paper ?

- You compared with SHE, claiming a much better efficiency ("latency reduction of 274x"), but you are comparing your network on MNIST, with their network on Penn Treebank. This is strange. Besides, if MNIST is the target, using a DNN SHE can reach 99.5% accuracy with a very small latency, largely outperforming your architecture. Actually, reaching 90.82% accuracy on MNIST is not impressive.

 - You use two NVIDIA A100 GPUs, while SHE [Lou & Jiang, 2019] used 10 CPU cores. How do you can you compare the latency and claim "latency reduction of 274x", when you seem to use very different computing capabilities ?

**Questions:**

- Why conducting experiments on an image dataset like MNIST, and not on a task more suited for RNN, like some time series ?

- you mention the Penn Treebank dataset used as one experiment in SHE, why not comparing on the same dataset ?

---

> ### Author Response · Authors · 2024-11-22
>
> Thank you for your detailed review and constructive feedback. Below, we address the weaknesses and questions you raised:
>
> **Weakness 1: Relation to the ICASSP paper**
> The ICASSP paper, which has been received positively, was tailored toward a specific use case: speaker identification. In contrast, this paper takes a broader perspective by focusing on evaluating large-scale RNNs in general. While the model in the ICASSP paper incorporates attention mechanisms, adding significant complexity, this work isolates the evaluation of large-scale, vanilla RNNs to explore the effectiveness of our quantization and OAR methods. Starting with a simpler dataset like MNIST and basic RNN model allows us to demonstrate the feasibility of our method while maintaining clarity in our analysis.
>
> **Weakness 2: Use of MNIST**
> The focus of this paper is not on optimizing performance on MNIST but on validating whether large-scale RNNs can be effectively evaluated with our quantization strategy and improved using OAR. MNIST, as a widely recognized benchmark, serves as a controlled starting point to establish the feasibility of our methods. By successfully applying our approach to large-scale RNNs, we provide a foundation for extending these techniques to more complex datasets and use cases in the future.
>
> **Weakness 3: Comparison with SHE and GPU use**
> SHE requires orders of magnitude more operations such as bootstrapping, even for basic tasks like addition. This makes it inherently slower, even with GPU acceleration. By leveraging GPUs in this work, we establish a benchmark for RNN evaluation over encrypted data, showcasing how GPUs can improve performance and make encrypted RNNs more practical. Our aim is to provide a valuable resource for future comparisons and research into RNN evaluation over encrypted data.
>
> **Questions: Why MNIST?**
> MNIST is widely used in privacy-preserving machine learning (PPML) studies as a benchmark to validate novel methods. By using MNIST, we align with research conventions, ensuring our results are comparable while providing an accessible demonstration of our method's effectiveness.

---

### Official Review · Reviewer_u5ZP · 2024-11-02

**Soundness:** 3
**Presentation:** 3
**Contribution:** 3
**Rating:** 6
**Confidence:** 4

**Summary:**

The paper presents a novel approach to enhancing the performance of recurrent neural networks (RNNs) on the MNIST dataset using Overflow-Aware Activity Regularization (OAR) to address overflow issues during quantization-aware training, particularly in low-bit-width settings. By employing fully homomorphic encryption (FHE) and GPU acceleration, the authors achieve over 90% accuracy with a latency of 2.1 seconds per sample, demonstrating significant improvements in both efficiency and model performance. The study highlights the effectiveness of OAR in maintaining accuracy during encrypted evaluations and suggests its potential for optimizing larger RNNs and future applications in privacy-preserving machine learning.

**Strengths:**

This work propose the method-Overflow-Aware Activity Regularization (OAR) offers several advantages over traditional L2 regularization in the context of RNNs, particularly when dealing with fully homomorphic encryption (FHE) for RNN inference:

1. **Mitigation of Numeric Overflow**: OAR specifically addresses the issue of numeric overflow that arises during computations with encrypted data. Traditional L2 regularization does not account for this overflow, which can lead to inaccuracies in the model's predictions. OAR helps maintain the integrity of the computations by managing the range of activations more effectively.

2. **Restoration of Accuracy**: While traditional L2 regularization may help prevent overfitting, it does not restore accuracy lost due to overflow effects. OAR has been shown to effectively recover lost accuracy in RNNs operating under FHE, demonstrating its efficacy in maintaining model performance even in challenging conditions.

3. **Efficiency in Encrypted Inference**: OAR allows for the use of a single ciphertext representation for inputs and activations, which reduces inference latency. This is particularly beneficial for RNNs, where low latency is crucial. Traditional L2 regularization does not provide this efficiency advantage in the context of encrypted data.

4. **Scalability**: OAR enables the scaling of RNNs to larger models without the exponential increase in computation that can occur with traditional methods that do not consider overflow. This scalability is essential for practical applications of RNNs in privacy-preserving settings.

In summary, OAR not only addresses the specific challenges posed by FHE but also enhances the overall performance and efficiency of RNNs compared to traditional L2 regularization.

**Weaknesses:**

As model compression is also proved to be effective in FHE inference. Model Compression: Implementing model compression techniques, such as pruning or knowledge distillation, could reduce the size of the RNN model, making it more efficient to evaluate under FHE. Smaller models require fewer resources for encryption and decryption, which can lead to faster inference times.

How does the proposed method to be incorporated with model compression here?

**Questions:**

See Weakness

---

> ### Author Response · Authors · 2024-11-22
>
> Thank you for your thoughtful review and positive feedback on our work! We appreciate your comments and the opportunity to address your question regarding model compression.
>
> Model compression indeed helps reduce model size. As discussed in lines 838–847, the ternarization of weights in our quantization strategy effectively acts as a form of pruning, as many weights are mapped to zero. This can be viewed as a type of compression integrated into our approach. This allows us to reduce the number of multiplications in the matrix-multiplications and increase efficiency as a result. We agree that exploring techniques like knowledge distillation would be a valuable direction for future work and are eager to experiment with it.
>
> Thank you again for your valuable insights!

---

### Official Review · Reviewer_qHBX · 2024-11-03

**Soundness:** 2
**Presentation:** 3
**Contribution:** 2
**Rating:** 3
**Confidence:** 4

**Summary:**

This paper introduces overflow-aware activity regularization (OAR), a regularization technique to mitigate numerical overflow during private RNN inference. The proposed method also allows for more efficient encryption parameters. This work enables large-scale private RNN inference. Experiments show that evaluating an RNN of 1.9M on one encrypted data sample from the MNIST dataset takes only 2.1 seconds.

**Strengths:**

1. The paper is well organized and easy to read. The key problem is explained clearly, i.e., the pre-activations can overflow in the message space.
2. The proposed OAR can effectively mitigate numerical overflow when the bit-width varies from 5 to 8.
3. The authors provided the code as well as details about their implementation.

**Weaknesses:**

1. The proposed OAR is not complete. The proposed OAR is largely built upon the anonymous reference in the supplementary materials. Specifically, the authors propose to add the OAR to the 4-step quantization procedure in the anonymous reference. Yet, without the OAR, the anonymous reference is said to have an accuracy drop of 25% (line 190). Thus, the practicality of the anonymous reference is questionable as well. The proposed OAR and the anonymous reference might be combined to offer better technical completeness.
2. The motivation is questionable. As shown in Table 1, when the RNN is quantized to 7-bit, the accuracy is 95.15%, despite the OAR metric being low. However, with the proposed OAR, the accuracy is even 1.18% lower while the OAR metric is higher. This indicates that enlarging the percentage of pre-activations positioned in the correct regions does not necessarily contribute to better performance, as neural networks are error-tolerant.
3. The novelty of the paper is limited. While the observation the authors put forward is sharp, the main contribution of the paper is the new regularization term introduced in Section 3.1, which optimizes the percentage of pre-activations values in the correct regions.
4. There is a lack of comprehensive evaluation. While the authors present some ablation studies of the proposed OAR, the overall evaluation is not comprehensive. There is a lack of end-to-end comparison with existing methods, such as SHE (Neurips'19) and ones based on the CKKS scheme.

**Questions:**

1. In Table 1, how does the latency change across different bit-widths?
2. Is the input binarized by the client before encryption (line 202)? If so, the description of the client (line 54) is not precise, as pre-encryption quantization of the input is not always needed, especially when the CKKS scheme is used.

---

> ### Author Response · Authors · 2024-11-22
>
> Thank you for your thoughtful feedback and detailed questions. We appreciate the opportunity to address your concerns. Below, we clarify and expand on the points you raised:
>
> 1. **Weakness 1: The proposed OAR is not complete**
>    We are unclear on what aspects of OAR you find incomplete. As noted in line 190, the baseline approach experiences a 25% accuracy drop without OAR, which directly motivated our work. OAR integrates into the four-step quantization procedure to address overflow effects, enhancing encrypted performance and preserving accuracy. Could you please provide further clarification on this comment so we can address it comprehensively?
>
> 2. **Weakness 2: The motivation is questionable**
>    Table 1 highlights OAR's substantial impact on accuracy, particularly for lower bit-widths. For instance, at bit-widths of 5 and 6, OAR improves accuracy by 70.85% and 42.53%, respectively. While accuracy for bit-width 7 decreases slightly (1.18%), this is likely due to overfitting during training (1000 epochs), where the model already had high accuracy without OAR application. These results suggest OAR is most impactful within certain range of bit-widths. Furthermore, this demonstrates the potential benefit of a hyperparameter search to fine-tune OAR's application, which we employed.
>
> 3. **Weakness 3: The novelty of the paper is limited**
>    We respectfully disagree. Our work tackles a critical challenge in privacy-preserving machine learning (PPML): addressing numerical overflow in RNN evaluation over CGGI-encrypted data. Previous methods, such as SHE, rely on computationally expensive operations like splitting of values into several ciphertexts, while our OAR-based approach maintains performance with a single ciphertext, significantly improving efficiency. This unique contribution represents an important step towards scalable and performant encrypted RNNs.
>
> 4. **Weakness 4: Limited comparability with prior work**
>    The structure and evaluation methodology of the RNN in SHE fundamentally differ from our approach, making direct comparisons inherently unfair. SHE relies on a multi-ciphertext representation and a fixed-point quantization scheme, resulting in significant computational complexity and latency. In contrast, our method employs a single-ciphertext representation and an innovative overflow-aware quantization strategy, directly addressing computational challenges in encrypted RNNs. These methodological differences make SHE’s performance metrics incomperable with our work. Similarly, evaluations using CKKS deviate from true RNN functionality, as they partition timesteps and reset the hidden state, effectively removing the ability to model long-range dependencies—an essential characteristic of RNNs. Thus, no meaningful comparison can be made. Our work instead builds upon the quantization strategy of the anonymous reference, overcoming its performance limitations while preserving its efficiency gains, offering a distinct and novel contribution to scalable encrypted RNNs. Further details are provided in Section 5 of the paper.
>
> 5. **Question 1: Latency across bit-widths (Table 1)**
>    The latency remains consistent across all bit-widths due to our single-ciphertext representation. Table 1 primarily demonstrates how OAR improves accuracy at various bit-widths. We can clarify this intent in the revised manuscript.
>
> 6. **Question 2: Input binarization before encryption (line 202)**
>    Yes, the client binarizes the input data before encryption. However, our main point in that sentence is to emphasize that the evaluation of the model does not require any back-and-forth communication of ciphertexts between the client and server during the inference process. Some existing methods rely on multi-party protocols to perform operations like re-encryption of ciphertexts because bootstrapping can be too expensive during inference. In contrast, our approach does not result in any interaction between the client and server once inference begins. From the first layer to the last, the server executes the entire inference independently. We can further clarify this point in the manuscript.

---

> > ### Comment · Reviewer_qHBX · 2024-11-23
> >
> > Thanks to the author for the response. I have comments as follows.
> >
> > * If I understand it correctly, the proposed OAR is a key component to make the four-step quantization in the anonymous reference practical. I still struggle to see how the contribution of OAR and the four-step quantization should be separated. On the one hand, without OAR, the four-step quantization in the anonymous is not practical (i.e., $25\\%$ accuracy drop). On the other hand, without the four-step quantization, the proposed OAR might not be necessary. For example, SHE does not need OAR to maintain high accuracy. Only when combined together the four-step quantization and OAR might serve as a complete solution for private RNN inference based on the CGGI scheme. The current statements in the manuscript, such as "the OAR is employed in the fourth step of the 4-step quantization procedure" (line 253), make it unclear how OAR can be decoupled from the four-step quantization. If OAR is a general technique that can operate without the four-step quantization, the authors should claim this clearly and add more results to support this claim.
> > * I understand that the poor performance (i.e., $25\\%$ accuracy drop) of the baseline four-step quantization motivates the authors to explore the OAR technique. However, the accuracy is high without OAR when the bit-width is 8 or 7. At the same time, as the authors mentioned in their answer to question 1, the latency is the same across all bit-widths. In this context, what is the motivation of applying OAR for 5-bit or 6-bit quantization? Also, the proposed OAR only works with 5-bit and 6-bit. It decreases the accuracy for 7-bit and 8-bit and does not apply to 3-bit or 4-bit.
> > * The authors seem to suggest that prior works could not achieve the true RNN functionality, so no comparison could be made. I still struggle to understand why "no meaningful comparison can be made" with related works. At least the runtime and accuracy are two general metrics for evaluating the performance of privacy-preserving machine learning methods. Comparing these metrics with prior works using the same dataset on the same hardware can better demonstrate the contribution of the proposed methods. On the other hand, I understand [1] deviates from the standard RNN functionality. However, in addition to [1], there are other works exploring private RNN inference using CKKS [2]. Although [2] uses a MKHE in the multi-party setting, the interactive bootstrapping can be adapted to the non-interactive one in the client-server setting. Additionally, since HE is not the only cryptographic primitive for secure RNN inference, discussion with works like [3] is also needed.
> >
> > I agree with Reviewer 4xG5 and think that the proposed OAR should be combined with the four-step quantization to offer better completeness.
> >
> > [1] Podschwadt et al., "Non-interactive privacy preserving recurrent neural network prediction with homomorphic encryption" CLOUD 2021.
> >
> > [2] Sav et al., "Privacy-Preserving Federated Recurrent Neural Networks" PoPETs 2023.
> >
> > [3] Rathee et al., "SIRNN: A Math Library for Secure RNN Inference" S&P 2021.

---

> > > ### Author Response · Authors · 2024-12-01
> > >
> > > Thank you for your response. We will try to address your comments in the following points:
> > >
> > > * Yes, our current experimentation demonstrates that the OAR method is closely integrated with the four-step quantization method presented in the attached ICASSP paper. However, the referenced paper focuses on a specific use case: speaker identification. The quantization method introduced there provides a means of quantizing large-scale RNNs and achieves a level of performance that may not be the greatest, but is the best with respect to the arithmetic that must be satisfied over encrypted data. Notably, **no other method achieves the same level of performance and efficiency attributed to the use of a single ciphertext representation aside from the four-step quantization method for RNNs**. The purpose of this quantization method is to eliminate the need for multiple ciphertexts to represent a single value, which significantly increases computational complexity and is impractical for latency-sensitive tasks like RNNs. We need to advance towards better, more efficient implementations. This is why our paper submitted to ICLR introduces the OAR method—work conducted after the ICASSP paper—to recover lost performance. The goals and scopes of each paper are independent, which is why they have not been combined. The other paper also explores attention mechanisms, which is not a focus of our ICLR paper. We aim to make it clear, as we explicitly mention in the title of the paper, that we are introducing methods that allow us to move **towards evaluating large-scale RNNs over encrypted data**. The methods presented in the paper, such as OAR and its integration into the four-step quantization method, provide a pathway that did not previously exist. This is why we strongly believe it is worth showcasing to the wider community.
> > >
> > > * In Section 4.1, as presented in Table 1, the evaluations are **performed over plaintext data**. The evaluations involving encrypted data are exclusively detailed in Section 4.2. Although we aimed in Section 4.1 to simulate a setting analogous to encrypted data—by employing the ModSign function and quantized values—there is a critical distinction: **we did not introduce noise into the plaintexts to mimic ciphertexts**. As discussed in Section 2.1, each ciphertext inherently contains noise as a security feature of the FHE scheme. This inherent noise becomes particularly problematic when operating within a 7-bit or 8-bit space; as noted in Section 2.2, the CGGI scheme is limited to a maximum message space of 7 or 8 bits. Maintaining the same RLWE dimension results in decreased redundancy in the lookup tables, leading to smaller rounding regions. Consequently, errors have a more pronounced effect, causing values at the extremes of the message space to shift more readily between positive and negative. Therefore, **the evaluation over encrypted data differs significantly from that over plaintext data at the 7-bit and 8-bit levels**. It is thus imperative to focus on the 5-bit and 6-bit settings, where the lookup tables have greater redundancy and larger rounding regions, and where errors have a significantly reduced impact on the evaluation compared to the 7-bit and 8-bit settings. This effect is evident in Table 4, where the percentage difference and Mean Absolute Error (MAE) between plaintext and encrypted pre-activations and activations, respectively, are minimal for each layer. This observation gives us confidence that errors do not substantially affect our network at the 5-bit and 6-bit levels.
> > >
> > > * The reference you provided [2] discusses the training of RNNs in a **federated learning context**, which significantly differs from the scope of our paper. Our focus is on evaluating the **inference of large-scale RNNs over encrypted data**, with an emphasis on latency reduction. Their evaluation involves RNNs with a very limited number of timesteps, whereas we aim to evaluate RNNs with an arbitrary number of timesteps. As shown in Table 4, our approach demonstrates that the error does not increase between layers. In contrast, **in CKKS, even when bootstrapping is employed, additional levels are added without refreshing the noise in the ciphertexts**. Therefore, the CKKS approach to evaluating arbitrary-length RNNs can be hindered by significant error growth that is not mitigated by bootstrapping, rendering it an unviable method towards large-scale RNN evaluation over encrypted data. Regarding your comment about reference [3], that paper discusses two-party computation (2PC) and interactive protocols, which represent a completely different paradigm and scope. Our paper focuses on the non-interactive evaluation of RNNs over encrypted data using fully homomorphic encryption. Therefore, we respectfully disagree that a comparison to that work is necessary, as the evaluation methods and scopes are entirely different.

---

### Meta-Review · Area_Chair_CmaT · 2024-12-21

**Metareview:**

The reviewers were split about this paper and did not come to a consensus: on one hand they appreciated the scalability and implementation details of the method, on the other they had concerns with (a) limited experimental results, (b) comparison with related work, and (c) the fact that the paper relies heavily on another unpublished paper that was shared to reviewers as an accompanying file. Two reviewers responded to the author feedback (qHBX and 4xG5, both with detailed discussions). No reviewers engaged in further discussion of the paper. After going through the paper and the discussion I have decided to vote to reject based on the above issues. Specifically, for (a) nearly all reviewers wanted to see a more extensive evaluation beyond MNIST. The authors responded by arguing that the the choice of MNIST aligned with research conventions and that this paper was designed as a proof of concept. I don’t buy these arguments: MNIST was introduced in 1998 and has been seen as outdated for at least the past 10 years. Further, given that the method works for a dataset that has 60k inputs and 784 features it is not a large lift to apply the method to additional RNN datasets. Finally, MNIST is rarely if ever used to benchmark RNN models. For (b) one reviewer pointed out that the authors had cited a “latency reduction of 274x” by comparing their network on MNIST with another method on Penn Treebank. This is not a fair comparison. The authors did not respond to this point. A reviewer also pointed out that 90.8% accuracy on MNIST is not impressive. The authors responded by arguing that this is a proof of concept. This response misses the main point that this is not a meaningful accuracy on MNIST: an SVM with a gaussian kernel achieved 98.6% test accuracy back in 1998 when the dataset was introduced [LeCun et al., 1998]. For (c) the reviewers found it problematic that the submission extensively leverages work from an unpublished paper. The main concern here is how to evaluate the contribution of the paper given its reliance on this other work. The authors argued that the other paper has a different, more complex use-case while this paper evaluates parts of the prior paper in a different setting. This  weakens the contribution of this paper down to evaluation: how do the quantization and OAR methods work for RNNs. Given that the evaluation is extremely limited, this harms the case for the paper. Given all of the above, I believe this work should be rejected at this time. Once these things and other issues mentioned in the reviews are addressed in an updated version, the work will be much improved.

**Additional Comments On Reviewer Discussion:**

See above details on this in the metareview.

---

### Decision · Program_Chairs · 2025-01-22

Reject